# *DRD4, DRD2, DAT1,* and *ANKK1* Genes Polymorphisms in Patients with Dual Diagnosis of Polysubstance Addictions

**DOI:** 10.3390/jcm9113593

**Published:** 2020-11-08

**Authors:** Jolanta Masiak, Jolanta Chmielowiec, Krzysztof Chmielowiec, Anna Grzywacz

**Affiliations:** 1Neurophysiological Independent Unit, Department of Psychiatry, Medical University of Lublin, 20-093 Lublin, Poland; jolantamasiak@wp.pl; 2Department of Hygiene and Epidemiology, Collegium Medicum, University of Zielona Góra, 65-046 Zielona Góra, Poland; chmiele1@o2.pl (J.C.); chmiele@vp.pl (K.C.); 3Independent Laboratory of Health Promotion, Pomeranian Medical University in Szczecin, 70-204 Szczecin, Poland

**Keywords:** dual diagnosis, polysubstance addictions, gene polymorphisms

## Abstract

**Background:** Approximately 25–50% of people diagnosed with substance use disorder experience psychiatric disorders, and this percentage is even higher if subclinical psychopathological symptomatology is taken into consideration. ”Dual diagnosis” implies the comorbidity of two disorders (mental disorder and addiction), but in a clinical setting, numerous dual diagnoses involve multiple addictions (polysubstance use means the concurrent use of more than one psychoactive substance). Clinical observations and epidemiological studies showed that the use of stimulants in combination with other substances results in additional risks. Apart from the clinical significance of the specificity of stimulants used in combination with other substances, only non-exhaustive research on the specificity of this comorbidity has been performed to date. The aim of the study was to analyze polymorphisms of the genes (DRD4 VNTR in exon III Ex3, DRD2 rs1076560, rs1800498, rs1079597, rs6276, as well as in the PROM promoter region (rs1799732, ANKK1 Tag1A rs1800497, DAT) in a group of patients diagnosed with polysubstance use disorder, including addiction to stimulants, and the co-occurrence of specific mental disorders in a group of patients diagnosed with polysubstance use disorder, including addiction to stimulants, compared to the group of patients diagnosed with polysubstance use disorder. **Methods:** The study group consisted of 601 male volunteers with psychoactive substance dependence (*n* = 300) and non-dependent controls (*n* = 301). The genomic DNA was extracted from venous blood using standard procedures. Genotyping was conducted with the real-time PCR method. All computations were performed using STATISTICA 13. **Results:** Psychotic disorders were significantly more common in the group of males with polysubstance addiction, including addiction to stimulants, compared to the group of males with polysubstance addiction without addiction to stimulants. In our own research, different statistical significances were found in the frequency of the DRD4 Ex3 gene polymorphism: s/s was more common in the study group. Psychotic disorders were more common in people addicted to stimulants compared to people addicted to other substances. **Conclusions:** In our study, psychotic disorders occurred more frequently in the study group of patients with polysubstance addiction, including addiction to stimulants, compared to the control group of patients with polysubstance addiction, but with no addiction to stimulants. Different statistical significances were found in the frequency of the DRD4 Ex3 gene polymorphism: s/s was more common in the study group, while the l/l genotype was less frequent in the study group. In DRD2 PROM rs 1799732, the del allele occurred more often than the ins allele in the study group. In the DRD4 Ex3 gene polymorphism, the s allele was more common in the study group, and the l allele was less frequent. In the DRD4 Ex3 gene polymorphism for the s/s genotype, psychotic disorders and generalized anxiety were more common, while for the s/l and l/l genotype, they were less frequent. The DRD4 Ex3 polymorphism s alleles were more common for depressive episode, dysthymia, and psychotic disorders as well as generalized anxiety disorder. We see a clear genetic aspect here. However, we want to be careful and draw no definite conclusions.

## 1. Introduction

In 2018, the dual diagnosis, i.e., co-occurrence of mental disorder and substance use disorders was found in approximately 9.2 million US adults aged 18 or older (3.7 percent of adults). In 2018, 3.2 million adults in the US experienced a comorbidity of substance use disorder as well as a serious mental illness, while in 2018, 11.4 million adults in the US were diagnosed with a serious mental illness [1]. Approximately 25–50% of people diagnosed with substance use disorder experience a mental disorder at the same time [2], and this percentage is even higher if subclinical psychopathological symptomatology is taken into consideration [3]. The results of The Epidemiologic Catchment Area Survey (ECA) reported that nearly 30% of patients with a psychiatric diagnosis suffered from substance use disorder. In addition, 48% of people diagnosed with schizophrenia, 55% of patients diagnosed with bipolar disorder, 90% of patients diagnosed with personality disorder, and more than 50% of patients with substance use disorder also developed a mental disorder during their lifetime [4]. Pre-existing mental disorders are significantly associated with an increased risk of developing substance use disorder related to alcohol, cannabis, as well as stimulants [5]. “Dual diagnosis” implies the comorbidity of two disorders (mental disorder and addiction), but in a clinical setting, numerous dual diagnoses involve multiple addictions (polysubstance use means the concurrent use of more than one psychoactive substance, or [6]) with one or more mental disorders consecutively [7]. As a result of the complexity of that multiple co-occurrence of psychiatric and substance use disorders, the concept of multimorbidity was formulated. Multimorbidity involves multiple mental disorders, substance use disorders, and general medical conditions [8]. These “complex” dual diagnoses present significant treatment challenges, with more severe illnesses and insufficiently integrated care for patients as well as a faster progression from regular use to substance use disorder [9,10]. Clinical observations and epidemiological studies showed that use of stimulants in combination with other substances results in additional risks. SAMHSA (Substance Abuse and Mental Health Services Administration) reports on increasing emergency department visits related to the use of stimulants, and 62% of the patients used stimulants with at least one more additional substance [11]. Apart from the clinical significance of the specificity of stimulants used in combination with other substances, inexhaustive research on the specificity of this comorbidity has been performed to date. One study showed that individuals with stimulant polysubstance use have a lower emotional empathy and a smaller social network compared to healthy controls [12]. The aim of the study was to analyze the polymorphisms of the genes (DRD4 Ex3, DRD2 (rs1076560, rs1800498, rs1079597, rs6276, rs1799732), ANKK1 Tag1A rs1800497, DAT1) and co-occurrence of specific mental disorders in the group of patients diagnosed with polysubstance use disorder, including stimulants, compared to the group of patients diagnosed with polysubstance use disorder. From the scientific point of view, it was also very interesting for us to learn about the specificity of co-occurring mental disorders in people who used stimulants in combination with other substances.

To advance the treatment of these complex conditions, more research is needed to reveal biological mechanisms of mental disorders and polysubstance addiction vulnerability [13,14]. Of course, we must bear in mind the importance of GWAS studies in the context of deliberations on addiction genetics and research methodology. The current studies on the brain-based linkages between these comorbidities are not intensive. Substance use disorder has a multifactorial etiopathology in which various factors are taken into consideration, including the individual genetic account for 40–60% of the susceptibility as well as environmental factors [15,16,17,18].

Attempts to understand these complex interactions are currently being made by means of analyzing, among other things, possible genetic factors that constitute the common background for this comorbidity. In the search for possible genetic risk factors related to substance use disorders, new molecular techniques are used to identify candidate genes involved in the regulation of neurotransmission and different processes modulated by dopamine [19]. It is well recognized that it is not only dopamine neurotransmission that is involved in substance use disorder. However, the role of dopamine transmission is unquestionable [20]. One of the candidate genes is the gene coding dopamine receptor 4 (DRD4). The DRD4 gene is located in chromosome 11p near the telomere, and it encodes the seven transmembrane G-protein coupled receptor, which responds to endogenous dopamine [21,22]. The variable number tandem repeat (VNTR) polymorphism occurs in exon III of the DRD4 gene. There is a 48 base pair sequence with a range of 2–11 repeats which manifests itself as either a “short” variant (five or fewer repeats—DRD4S) or a “long” variant (six or more repeats—DRD4L). The two, four, and seven repeats are considered as the most common genotypes [23]; the length of the variant has functional effects on the dopamine receptor. Repeats seven and more were correlated with blunted intracellular sensitivity and responsiveness to dopamine, which may contribute to the differences in motivation, sensation-seeking, and impulsivity often observed among carriers of DRD4S and DRD4L [24]. DRD4 exon III (VNTR) polymorphism was reported as a candidate genetic variant associated with substance use disorder (SUD) susceptibility in different populations [25] as well as a number of approach-oriented behavioral phenotypes and psychiatric disorders. Another candidate gene is the gene coding dopamine receptor 2 (DRD2). There is a continuing controversy concerning the role of the dopamine D2 receptor gene (DRD2) in association with alcohol use disorder (AUD) and other psychopathologies [26]. The research confirmed the role of dopaminergic transmission through the D2 receptor in addiction: it determines the expression of reward, diminishes alcohol consumption in animal studies, and has associations with vulnerability to addiction [27,28]. The gene of the dopamine receptor D2 (DRD2) is located in the chromosome 11q23 and spans 65,56 kilobase. The DRD2 gene includes eight exons that undergo transcription to messenger RNA of 2713 kb that is translated to 443 amino acid proteins. Skipping the sixth exon results in the formation of a short form of a receptor when the long variant of the receptor protein is constituted of 29 amino acids. The two isoforms of the D2 receptors have a different affinity with inhibitory G-proteins [29]. Polymorphic versions of the DRD2 gene rs1076560 located in its 6th intron are considered important factors in the genetics of mental disorders and behavior. The presence of the A allele of rs1076560 is associated with a lower expression of the short isoform relative to the long isoform in the prefrontal cortex and caudate putamen. A low activity of D2 receptors was observed in patients with alcohol dependence and cocaine and opiates abuse [30,31,32].

The aim of the study was to analyze polymorphisms of the genes (DRD4 Ex3, DRD2 (rs1076560, rs1800498, rs1079597, rs6276, rs1799732), ANKK1 Tag1A rs1800497, DAT1) and co-occurrence of specific mental disorders in the group of patients diagnosed with polysubstance use disorder including stimulants, compared to the group of patients diagnosed with polysubstance use disorder.

## 2. Experimental Section

### 2.1. Subjects

The study group consisted of 601 male volunteers with psychoactive substance dependence (*n* = 300; mean age = 28.18, SD = 6.45) and non-dependent controls (*n* = 301; mean age = 22; SD = 4.57). From the group of patients with psychoactive substance dependence, those dependent on stimulants (F15.2 *n* = 247; mean age = 27.6, SD = 5.75) and other psychoactive substances (*n* = 53; mean age = 31, SD = 8.52) were distinguished. Only men were included in the study, as it was a section of a larger study in which fluctuations in women’s sex hormone cycles may have affected the examined properties. For further analysis, a group of men dependent on many substances, including stimulants, were selected in order to achieve the aim of the study.

The psychiatric examination was also performed on the control group. The occurrence of mental disorders in that group was evaluated. The occurrence of any mental disorder in a candidate for the control group was a disqualifying criterion.

The percentage distribution of a particular type of addiction in the patients under study is shown in Table 1. After obtaining the approval of the Bioethics Committee of the Pomeranian Medical University in Szczecin (KB-0012/106/16) and an informed, written consent of the participants, the study was conducted in the Independent Laboratory of Health Promotion. Patients with psychoactive substance dependence were recruited after at least 3 months of abstinence in addiction treatment facilities. We did not differentiate the simultaneous co-ingestion of different substances from concurrent (different substances used on the same or separate occasions within the same period) polysubstance use.

The dependent patients and controls were clinically tested by psychiatrists for the following disorders: depressive episode, dysthymia, suicide attempt, hypo or manic episode, panic-related disorders, agoraphobia, social phobia, obsessive-compulsive disorder (OCD), post-traumatic stress disorder (PTSD), psychotic disorder, and generalized anxiety.

The history of dependence was collected using the Polish version of ICD-10, authors’ survey, and the medical history. DNA was provided from the venous blood.

### 2.2. Genotyping

The genomic DNA was extracted from venous blood using standard procedures. Genotyping was conducted with the real-time PCR method.

The LightCycler^®^ 480 II System (Roche Diagnostic, Basel, Switzerland) was applied to perform the fluorescence resonance energy into the genotypic data. The data related to the DRD2 gene polymorphism were obtained under the following conditions: PCR was performed with 50 ng DNA of each sample in a final volume of 20 μl containing 2 µl reaction mix, 0.5 mM of each primer, 0.2 mM of each hybridization probe, and 2 mM of MgCl2 according to the manufacturer’s instructions with initial denaturation (95 °C for 10 min) and then 35 cycles of denaturation (95 °C for 10 s), annealing (60 °C for 10 s), and extension (72 °C for 15 s). After amplification, a melting curve was generated by holding the reaction at 40 °C for 20 s and then heating slowly to a level of 95 °C. The fluorescence signal was plotted against temperature to provide melting curves for each sample.

The peaks for rs1800497 were obtained at 58.95 °C for the T allele and 67.17 °C for the C allele. For rs6276, they were at 59.14 °C for the G allele and at 67.66 °C for the A allele. For rs1076560, the peaks were obtained at 57.13 °C for the A allele and 64.40 °C for the C allele. For rs1800498, the peaks were obtained at 57.87 °C for the T allele and 66.34 °C for the C allele. For rs1079597, the peaks were obtained at 57.41 °C for the G allele and 62.25 °C for the A allele. For ANKK1 rs1800497, they were obtained at 58.95 °C for the T (A1) allele and at 67.17 °C for the °C (A2) allele.

The DAT1 genotypes were grouped according to the presence of nine or 10 repeat variants. Genotyping was performed using the PCR-VNTR method, with primers: F: 50-TGT GGT GTA GGG AAC GGC CTG Ag 30, R: 50-CTT CCT GGA GGT CAC GGC TCA AGG 30; in the final volume of 25 L PCR mix per reaction, with l00 ng genomic DNA, 10 pmol of primers, 50 mM KCl, 10 mM TrisHCl, 1.5 mM MgCl2, 200 M dATP, dCTP, dTTP, dGTP, and 0.8 U of the Tag polymerase. The conditions for the reaction were as follows: 5 min of initial denaturation in 94 °C, 55 s of denaturation in 94 °C, 50 s of primer hybridization in 55 °C, and 1 min of elongation in 72 °C, repeated in 30 cycles, with 10 min of final elongation in 72 °C. The amplified products were visualized using ethidium bromide stained gel electrophoresis (3% agarose) and UV photography. The products were 450 bp in length for 10 repeat alleles and 410 bp for nine repeat alleles.

The DRD4 genotypes were grouped based on the presence of the short (2–5 repeat) and long (6–11 repeat) variants. Genotyping was performed using the PCR-VNTR method with the following primers: F: 50-GCG ACT ACG TGG TCT ACT CG 3 0, R: 50-AGG ACC CTC ATG GCC TTG 3 0; in the final volume of 25 μL PCR mix per reaction, with l00 ng genomic DNA, 10 pmol of primers, 50 mM KCl, 10 mM TrisHCl, 1.5 mM MgCl2, 200 μM dATP, dCTP, dTTP, dGTP and 0.8 U of the Tag polymerase. The conditions for the reaction: 3 min of initial denaturation in 95 °C, cycling 30 s of denaturation in 95 °C, 1 min of primers hybridization in 63 °C and 30 s of elongation in 72 °C, repeated in 35 cycles, 5 min of final elongation in 72 °C. The amplified products were visualized using ethidium bromide stained gel electrophoresis (3% agarose) and UV photography. The products ranged from 379 bp (2 repeats) to 811 (11 repeats). The products were divided into two groups: short alleles (S, 2–5 repeats) and long alleles (L, 6–11 repeats).

### 2.3. Statistical Analysis

The relations between DRD4 Ex3, DRD2 rs1076560, DRD2 Tag1D rs1800498, DRD2Tag1B rs1079597, DRD2 Ex8 rs6276, DRD2 PROM. rs1799732, ANKK1 Tag1A rs1800497, DAT1 variants, in control subjects with dependence on stimulants and the occurrence of mental disorders were tested with the chi square test. No statistically significant associations were found between the polymorphism of the DRD2 rs1076560, DRD2 Tag1D rs1800498, DRD2Tag1B rs1079597, DRD2 Ex8 rs6276, DRD2 PROM.rs1799732, ANKK1 Tag1A rs1800497, DAT genes, and psychiatric disorders in patients addicted to stimulants and in the control group. In the study group, further analysis of the relationship between mental disorders was performed only for the DRD4 Ex3 gene polymorphism. In that case, the chi square test was also used. For these variables, the Bonferroni multiple comparisons correction was applied, and the accepted level of significance was 0.0045 (0.05/11). All computations were performed using STATISTICA 13 (Tibco Software Inc, Palo Alto, CA, USA) for Windows (Microsoft Corporation, Redmond, WA, USA).

## 3. Results

Significant differences were found in the DRD4 Ex3 polymorphism for addiction-stimulating substances and the control group genotypes (s/l 0.31 vs s/l 0.33, s/s 0.65 vs. s/s 0.59, l/l 0.04 vs. l/l 0.09, χ^2^ = 6.27, *p* = 0.043) and the frequency of DRD4 Ex3 alleles (s 0.81 vs. s 0.75, l 0.19 vs. l 0.25, χ^2^ = 5.05, *p* = 0.025). Statistically significant differences were also found only in the frequency of the DRD2 PROM allele rs1799732 between addiction-stimulating substances and control groups (del 0.15 vs. del 0.11, ins 0.85 vs. ins 0.89, χ^2^ = 5.07, *p* = 0.024). For other gene polymorphisms (DRD2 rs1076560, DRD2 Tag1D rs1800498, DRD2Tag1B rs1079597, DRD2 Ex8 rs6276, ANKK1 Tag1A rs1800497, DAT1) and their allele distribution, no significant statistical differences were found (Table 2).

By comparing the frequency of occurrence of particular mental disorders between people addicted to stimulants and people dependent on other psychoactive substances, statistically significant differences were shown only in the frequency of psychotic disorders. Psychotic disorders occurred more frequently in people addicted to stimulants (0.49 vs. 0.22, χ^2^ = 13.24, *p* = 0.0003, Table 3).

A relationship was found between the presence or absence of psychotic disorders (the study group and the control group in total *n* = 601) and the polymorphism of the DRD4 Ex3 genotype (s/l 0.25 vs. s/l 0.34, s/s 0.72 vs. s/s 0.59, l/l 0.03 vs. l/l 0.07, χ^2^ = 7.19, *p* = 0.027) and the frequency of DRD4 Ex3 alleles (s 0.84 vs. s 0.76, l 0.16 vs. l 0.24, χ^2^ = 7.72, *p* = 0.006). A relationship was also found between the presence of generalized anxiety or its absence and polymorphism of the DRD4 Ex3 genotype (s/l 0.22 vs. s/l 0.33, s/s 0.78 vs. s/s 0.59, l/l 0.00 vs. l/l 0.07, χ^2^ = 10.57, *p* = 0.005) and the frequency of DRD4 Ex3 alleles (s 0.89 vs. s 0.76, l 0.11 vs. l 0.24, χ^2^ = 11.40, *p* = 0.0007). However, only significant differences were found for the frequency of the DRD4 Ex3 allele in people with a diagnosed depressive episode (s 0.84 vs. s 0.77, l 0.16 vs. l 0.23, χ^2^ = 4.02, *p* = 0.045) and dysthymia (s 0.86 vs. s 0.77, l 0.14 vs. l 0.23, χ^2^ = 4.71, *p* = 0.03) compared to people without those mental disorders. After the Bonferroni correction was applied, a relationship was also found between the presence of generalized anxiety or its absence and the polymorphism of the DRD4 Ex3 alleles (Table 4).

## 4. Discussion

The initial consideration in our research was an assessment of the prevalence of certain polymorphisms in candidate genes in people addicted to multiple substances, including stimulants. We also investigated the presence of psychiatric disorders in people addicted to multiple substances, including stimulants, versus people addicted to multiple psychoactive substances.

An additional factor conditioning the clinical picture is the type of substance used by the patient. In our opinion, the most interesting result of the analyses is that psychotic disorders occurred more frequently in the study group of patients with polysubstance addiction, including addiction to stimulants, compared to the control group of patients with polysubstance addiction, but without addiction to stimulants (Table 3). Stimulants cause hallucinations and a delusional interpretation of reality, even in mentally healthy people. Stimulants are among the most popular psychoactive drugs used by people diagnosed with psychosis. Stimulants may cause psychotic states similar to schizophrenia in mentally healthy people or exacerbate symptoms of pre-existing psychoses. The symptoms include paranoid states and may recur in the form of flashbacks after long periods of abstinence [33,34].

Following the literature and the statement of “common susceptibility”, for our analyses, we chose polymorphisms of genes associated with dopaminergic receptors and transporters. A significantly increased prevalence of substance use disorders in people diagnosed with psychotic disorders compared to general population is well confirmed by research. Theories explaining the comorbidity of addiction and schizophrenia include the primary addiction or ”shared vulnerability” hypothesis of shared genetic and environmental risk factors and neurobiological dysfunctions within the meso-cortico-limbic dopamine system, which predisposes to schizophrenia but also to substance use disorder [35]. This hypothesis proposes that the genetic determinants of risk for the occurrence of schizophrenia predispose to substance use disorder which, in turn, serves as a risk factor for the development of schizophrenia symptomatology [36].

Genetic factors were studied in regard to susceptibility to the development of schizophrenia and co-occurring substance use disorder. It was also confirmed that polygenic risk scores for schizophrenia are associated with cannabis use, cocaine use, nicotine use, and severe alcohol use [37]. Since such a strong genetic correlation was found, our first analysis was justified.

In our own research, different statistical significances were found in the frequency of the DRD4 Ex3 gene polymorphism: s/s is more common in the study group, while the l/l genotype is less frequent in the study group (Table 2). On the other hand, in DRD2 PROM rs 1799732, the del allele occurs more often than the ins allele in the study group (Table 2). In the DRD4 Ex3 gene polymorphism, the s allele is more common in the study group, and the l allele is less frequent (Table 2). Such analysis shows us the genotypic and allelic characteristics in the study group. We found that in our group, some variants were statistically significantly more frequent, which confirmed the theory related to the aspect of a genetic component in addiction and pointed to which area of research should be explored. However, in our study, we wanted to get a broader picture of the group, not only in terms of genetic conditions, but mainly in terms of all differences related to clinical aspects of dual diagnosis. Table 3 shows an interesting aspect, where the difference at a level of statistical significance is shown. Specifically, psychotic disorders were more common in people addicted to stimulants compared to people addicted to other substances (Table 3). Interestingly, the literature includes reports on the widely described bipolar spectrum resulting from the use of stimulants. The bipolar spectrum was in fact the only profile that differentiated heroin users or people with alcohol dependence from healthy people [38,39].

In addition to research focused on addiction, other authors point to the use of stimulants (substances from the group of stimulants), possibly in combination with alcohol and cannabinoids, as characteristic of the bipolar spectrum. The concept of bipolarity resulting from the use of stimulants was proposed. The study shows that patients who had a significantly elevated mood in the bipolar spectrum had used stimulants for years before they developed more severe mood disorders, and the use of these stimulants resulted in a controlled and sustained subclinical rewarding mood condition. This was important both for the emergence of bipolar and dependence traits [40]. In our study group, no relationship of this kind was observed.

A similar study was carried out on a group of politoxicomaniac patients—where groups of patients with alcohol and heroin dependence, alcohol and cocaine dependence, and heroin and cocaine dependence were compared, respectively. The pattern of repeated use of alcohol is typical of people with bipolar disorder with a low intensity of mood swings [41,42,43,44,45]. Depressive disorders were not associated with any of the combinations. However, people addicted both to heroin and alcohol developed their addiction by skipping drug doses, ending treatment too early, or receiving insufficient pharmacological treatment, which appears to be a substitute for opioid use [46,47].

In our study, in the DRD4 Ex3 gene polymorphism for the s/s genotype, psychotic disorders and generalized anxiety were more common, while for the s/l and l/l genotype, they were less frequent (Table 4). We see a clear genetic aspect here. However, we want to be careful and draw no definite conclusions. The DRD4 Ex3 polymorphism s alleles were more common for depressive episode, dysthymia, psychotic disorders, and generalized anxiety disorder (Table 4).

Table 4 shows the results with Bonferroni’s correction; differences in the distribution of alleles in the study group were found to be significantly more frequent in patients with generalized anxiety in the study group (using many psychoactive substances, including stimulants) and healthy individuals from the control group. Analogically, allele l was less frequent in people with generalized anxiety in this group. These results may indicate that generalized anxiety is related to the polymorphisms of the DRD4 gene, but it was found that significant differences in individuals dependent on multiple substances, including stimulants, are not only found in the DRD4 gene but also in the DRD2 gene in the RS 1799732 promoter. The short variant of the DRD4 VNTR exon III was associated with increased neurotic symptoms in healthy individuals [48].

In the promoter region of DRD2 rs1799732 in the group of patients diagnosed with polysubstance use disorder, including addiction to stimulants, allel del ins occurred significantly more often compared to the group of healthy controls, and allel ins occurred less often. It may be assumed that the more frequent occurrence of the s allele is connected with the more frequent occurrence of generalized anxiety and similarly, the s allele is more frequent in people dependent on many substances, including stimulants. Therefore, it is worth considering whether the presence of the s allele is not associated with greater susceptibility to mental disorders, such as generalized anxiety disorder and dependence on many substances, including stimulants.

Substance use disorders are also heterogeneous, since not all clinical images correspond to a chronic, recurrent loss of control over use (dependence). In addition, not all cases of multiple use have the same dynamics for primary dependence and comorbid psychiatric disorders; therefore, the biological aspect should also be considered. The biology of addictions is still unknown and based on the research we have carried out, we are aware of the need for further analyses. In the case of multigeneity and multifactoriality in addiction, the GWAS seems to be the most accurate analysis.

Large-scale GWAS are well powered to detect genetic effects in or near candidate genes, and their failure to implicate candidate genes—while implicating many other loci—is informative. Most promisingly, what has emerged (and is still emerging) from GWAS is a set of novel variants that provide clues to the etiology of psychiatric diseases [14]. GWAS will be the next stage of our research when gathering the appropriate group size. The deviation from the classical criteria of intoxication and particular substance withdrawal syndrome inspires performing a dual diagnosis.

## 5. Conclusions

In studies on addiction, we should be particularly sensitive to the criterion of dual diagnosis. The combination of different substances used simultaneously has a diagnostic and therapeutic significance as well. Moreover, the factors related to differentiating patients may lie in the biological aspect, e.g., in the differentiation of individual polymorphic variants of candidate genes. In our study, psychotic disorders occurred more frequently in the study group of patients with polysubstance addiction, including addiction to stimulants, compared to the control group of patients with polysubstance addiction, but without addiction to stimulants. Different statistical significances were found in the frequency of the DRD4 Ex3 gene polymorphism: s/s was more common in the study group of patients addicted to stimulants and other psychoactive substances, while the l/l genotype was less frequent in the study group. In DRD2 PROM rs 1799732, the del allele occured more often than the ins allele in the study group. In the DRD4 Ex3 gene polymorphism, the s allele was more common in the study group, and the l allele was less frequent. In the DRD4 Ex3 gene polymorphism for the s/s genotype, psychotic disorders and generalized anxiety were more common, while for the s/l and l/l genotype, they were less frequent. The DRD4 Ex3 polymorphism s alleles were more common for depressive episode, dysthymia, psychotic disorders, and generalized anxiety disorder. We see a clear genetic aspect here. However, we want to be careful and draw no definite conclusions.

## Figures and Tables

**Table 1 jcm-09-03593-t001:** Type of use of psychoactive substances in addicts.

Type of Substance/Addiction Used	All Addicted (*n* = 300)		Addicted to Stimulants (*n* = 247)		Addicted to Other Psychoactive Substances (*n* = 53)	
	*n*	%	*n*	%	*n*	%
Behavioral addiction	128	43	107	43	21	40
Designer drugs	73	24	56	23	17	32
F10.2-alcohol	166	55	134	54	32	60
F11.2-opiates	61	20	44	18	17	32
F12.2-cannabinols	214	71	181	73	33	62
F13.2-sedatives and hypnotics	38	13	22	9	16	30
F14.2-cocaine	31	10	29	12	2	4
F15.2-stimulants	247	82	247	100	-	-
F16.2-hallucinogenic	31	10	31	13	0	0
F19.2-mixed addictions	172	57	156	63	16	30

The total is not 100%. It was found that the addicts used various psychoactive substances.

**Table 2 jcm-09-03593-t002:** Genetic polymorphism dopamine receptor (DRD4 Ex3), dopamine receptor 2 (DRD2) in addicts, and coexisting F15.2-stimulants.

	Genotype			Allele	
DRD2 rs1076560					
	C/C *n* (%)	A/C *n* (%)	A/A *n* (%)	C *n* (%)	A *n* (%)
Addiction-stimulating substances *n* = 247	160 (64.78%)	77 (31.17%)	10 (4.05%)	397 (80.36%)	97 (19.64%)
Control *n* = 301	208 (69.10%)	82 (27.24%)	11 (3.65%)	498 (82.72%)	104 (17.28%)
Pearson’s χ^2^ (*p* value)	1.155 (0.561)			1.010 (0.315)	
DRD2 Tag1D rs1800498					
	T/T *n* (%)	C/T *n* (%)	C/C *n* (%)	T *n* (%)	C *n* (%)
Addiction-stimulating substances *n* = 247	77 (31.17%)	118 (47.77%)	52 (21.05%)	272 (55.06%)	222 (44.94%)
Control *n* = 301	108 (35.88%)	142 (47.18%)	51 (16.94%)	358 (59.47%)	244 (40.53%)
Pearson’s χ^2^ *p* value	2.119 (0.347)			2.160 (0.142)	
DRD2Tag1B rs1079597					
	G/G *n* (%)	A/G *n* (%)	A/A *n* (%)	G *n* (%)	A *n* (%)
Addiction-stimulating substances *n* = 247	165 (66.80%)	74 (29.96%)	8 (3.24%)	404 (81.78%)	90 (18.22%)
Control *n* = 301	207 (68.77%)	83 (27.57%)	11 (3.65%)	497 (82.56)	105 (17.44%)
Pearson’s χ^2^ *p* value	0.414 (0.813)			0.110 (0.738)	
DRD2 Ex8 rs6276					
	A/G *n* (%)	A/A *n* (%)	G/G *n* (%)	A *n* (%)	G *n* (%)
Addiction-stimulating substances *n* = 247	118 (47.77%)	100 (40.49%)	29 (11.74%)	336 (68.02%)	158 (31.98)
Control *n* = 301	129 (42.86%)	127 (42.19%)	45 (14.95%)	385 (63.95%)	217 (36.05%)
Pearson’s χ^2^ *p* value	1.857 (0.395)			1.990 (0.158)	
DRD2 PROM. rs1799732					
	del/del *n* (%)	ins/ins *n* (%)	ins/del *n* (%)	del *n* (%)	ins *n* (%)
Addiction-stimulating substances *n* = 247	9 (3.64%)	181 (73.28%)	57 (23.08%)	75 (15.18%)	419 (84.82)
Control *n* = 301	4 (1.33%)	241 (80.07%)	56 (18.60%)	64 (10.63%)	538 (89.37%)
Pearson’s χ^2^ *p* value	5.192 (0.074)			5.07* (0.024)	
ANKK1 Tag1A rs1800497					
	C/C *n* (%)	C/T *n* (%)	T/T *n* (%)	C *n* (%)	T *n* (%)
Addiction-stimulating substances *n* = 247	154 (62.35%)	82 (33.20%)	11 (4.45%)	390 (78.95%)	104 (21.05)
Control *n* = 301	199 (66.33%)	95 (31.33%)	7 (2.33%)	493 (81.89%)	109 (18.11%)
Pearson’s χ^2^ *p* value	2.330 (0.312)			1.500 (0.220)	
DAT1					
	9/10 *n* (%)	9/9 *n* (%)	10/10 *n* (%)	9 *n* (%)	10 *n* (%)
Addiction-stimulating substances *n* = 247	101 (40.89%)	7 (2.83%)	139 (56.28%)	115 (23.28)	379 (76.72)
Control *n* = 301	114 (37.87%)	19 (6.31%)	168 (55.81%)	152 (25.25%)	450 (74.75%)
Pearson’s χ^2^ *p* value	3.779 (0.151)			0.570 (0.450)	
DRD4 Ex3					
	s/l *n* (%)	s/s *n* (%)	l/l *n* (%)	s *n* (%)	l *n* (%)
Addiction-stimulating substances *n* = 247	77 (31.17%)	161 (65.18%)	9 (3.64%)	399 (80.77%)	95 (19.23%)
Control *n* = 301	98 (32.56%)	177 (58.80%)	26 (8.64%)	452 (75.08%)	150 (24.92%)
Pearson’s χ^2^ *p* value	6.274* (0.043)			5.050* (0.025)	

* Significant statistical differences.

**Table 3 jcm-09-03593-t003:** Mental disorders in addicts and coexisting addiction F15.2-stimulants.

Mental Disorders	Addiction	Not *n* (%)	Yes *n* (%)	Pearson’s χ^2^ *p* Value
Depressive episode	other addictions *n* = 54	37 (68.52%)	17 (31.48%)	0.0002 (0.989)
	addiction-stimulating substances *n* = 247	169 (68.42%)	78 (31.58%)	
Dysthymia	other addictions *n* = 54	47 (87.04%)	7 (12.96%)	1.242 (0.265)
	addiction-stimulating substances *n* = 247	199 (80.57%)	48 (19.43%)	
Suicide attempts	other addictions *n* = 54	51 (94.44%)	3 (5.56%)	0.001 (0.974)
	addiction-stimulating substances *n* = 247	233 (94.33%)	14 (5.67%)	
Hypomanic or manic episode	other addictions *n* = 54	38 (70.37%)	16 (29.63%)	0.0001 (0.991)
	addiction-stimulating substances *n* = 247	174 (70.45%)	73 (29.55%)	
Panic-related disorder	other addictions *n* = 54	49 (90.74%)	5 (9.26%)	0.196 (0.658)
	addiction-stimulating substances *n* = 247	219 (88.66%)	28 (11.34%)	
Agoraphobia	other addictions *n* = 54	49 (90.74%)	5 (9.26%)	0.078 (0.779)
	addiction-stimulating substances *n* = 247	227 (91.90%)	20 8.10% ()	
Social phobia	other addictions *n* = 54	48 (88.89%)	6 (11.11%)	1.749 (0.186)
	addiction-stimulating substances *n* = 247	201 (81.38%)	46 (18.62%)	
OCD	other addictions *n* = 54	47 (87.04%)	7 (12.96%)	0.856 (0.355)
	addiction-stimulating substances *n* = 247	202 (81.78%)	45 (18.22%)	
PTSD	other addictions *n* = 54	50 (92.59%)	4 (7.41%)	0.029 (0.865)
	addiction-stimulating substances *n* = 247	227 (91.90%)	20 (8.10%)	
Psychotic disorders	other addictions *n* = 54	42 (77.78%)	12 (22.22%)	13.244*# (0.0003)
	addiction-stimulating substances *n* = 247	125 (50.61%)	122 (49.39%)	
Generalized anxiety	other addictions *n* = 54	44 (81.48%)	10 (18.52%)	1.440 (0.230)
	addiction-stimulating substances *n* = 247	182 (73.68%)	65 (26.32%)	

* Significant statistical differences. # Bonferroni correction was used, and the p value was reduced to 0.0045 (*p* = 0.05/11 (number of statistical tests conducted)).

**Table 4 jcm-09-03593-t004:** Polymorphism of the DRD4 Ex3 gene in the study group (people addicted to stimulants and the control group), including mental disorders.

	Genotype			Allele	
DRD4 Ex3					
	s/l *n* (%)	s/s *n* (%)	l/l *n* (%)	s *n* (%)	l *n* (%)
Depressive episode - not *n* = 468	153 (32.69%)	282 (60.26%)	33 (7.05%)	717 (76.60)	219 (23.40%)
Depressive episode - yes *n* = 80	22 (27.50%)	56 (70.00%)	2 (2.50%)	134 (83.75)	26 (16.25)
Pearson’s χ^2^ *p* value	3.844 (0.146)			4.020* (0.045)	
Dysthymia - not *n* = 500	164 (32.80%)	302 (60.40%)	34 (6.80%)	768 (76.80%)	232 (23.20%)
Dysthymia - yes *n* = 48	11 (22.92%)	36 (75.00%)	1 (2.08%)	83 (86.46%)	13 (13.54%)
Pearson’s χ^2^ *p* value	4.379 (0.112)			4.710* (0.030)	
Suicide attempts - not *n* = 535	171 (31.96%)	329 (61.50%)	35 (6.54%)	829 (77.48%)	241 (22.52%)
Suicide attempts - yes *n* = 13	4 (30.77%)	9 (69.23%)	0 (0.00%)	22 (84.62%)	4 (15.38%)
Pearson’s χ^2^ *p* value	0.979 (0.612)			0.750 (0.388)	
Hypo or manic episode - not *n* = 477	154 (32.29%)	290 (60.80%)	33 (6.92%)	734 (76.94%)	220 (23.06%)
Hypo or manic episode - yes *n* = 71	21 (29.58%)	48 (67.61%)	2 (2.82%)	117 (82.39%)	25 (17.61%)
Pearson’s χ^2^ *p* value	2.234 (0.327)			2.120 (0.145)	
Hypo or manic episode - not *n* = 520	167 (32.12%)	318 (61.15%)	35 (6.73%)	802 (77.21%)	237 (22.79%)
Hypo or manic episode - yes *n* = 28	8 (28.57%)	20 (71.43%)	0 (0.00%)	48 (85.71%)	8 (14.29%)
Pearson’s χ^2^ *p* value	2.444 (0.295)			2.220 (0.136)	
Agoraphobia - not *n* = 527	168 (31.88%)	324 (61.48%)	35 (6.64%)	816 (77.42%)	238 (22.58)
Agoraphobia - yes *n* = 21	7 (33.33%)	14 (66.67%)	0 (0.00%)	35 (83.33%)	7 (16.67%)
Pearson’s χ^2^ *p* value	1.496 (0.473)			0.810 (0.367)	
Social phobia - not *n* = 502	158 (31.47%)	309 (61.55%)	35 (6.97%)	776 (77.29%)	228 (22.71%)
Social phobia - yes *n* = 46	17 (36.96%)	29 (63.04%)	0 (0.00%)	75 (81.52%)	17 (18.48%)
Pearson’s χ^2^ *p* value	3.618 (0.163)			0.870 (0.351)	
OCD -not *n* = 503	164 (32.60%)	305 (60.64%)	34 (6.76%)	774 (76.94%)	232 (23.06%)
OCD - yes *n* = 45	11 (24.44%)	33 (73.33%)	1 (2.22%)	77 (85.56%)	13 (14.44%)
Pearson’s χ^2^ *p* value	3.272 (0.195)			3.530 (0.060)	
PTSD -not *n* = 528	169 (32.01%)	325 (61.55%)	34 (6.44%)	819 (77.56%)	237 (22.44%)
PTSD - yes *n* = 20	6 (30.00%)	13 (65.00%)	1 (5.00%)	32 (80.00%)	8 (20.00%)
Pearson’s χ^2^ *p* value	0.123 (0.939)			0.130 (0.716)	
Psychotic disorder - not *n* = 425	144 (33.88%)	250 (58.82%)	31 (7.29%)	644 (75.76%)	206 (24.23%)
Psychotic disorder - yes *n* = 123	31 (25.20%)	88 (71.54%)	4 (3.25%)	207 (84.15%)	39 (15.85%)
Pearson’s χ^2^ *p* value	7.193* (0.027)			7.720* (0.006)	
Generalized anxiety - not *n* = 483	161 (33.33%)	287 (59.42%)	35 (7.25%)	735 (76.09%)	231 (23.91%)
Generalized anxiety - yes *n* = 65	14 (21.54%)	51 (78.46%)	0 (0.00%)	116 (89.23%)	14 (10.77%)
Pearson’s χ^2^ *p* value	10.573* (0.005)			11.400*# (0.0007)	

* Significant statistical differences. # Bonferroni correction was used, and the *p* value was reduced to 0.0045 (*p* = 0.05/11 (number of statistical tests conducted)).

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
