# Peer review of "DRD4, DRD2, DAT1, and ANKK1 Genes Polymorphisms in Patients with Dual Diagnosis of Polysubstance Addictions"

_jcm, 2020, doi:10.3390/jcm9113593_

Round 1

Reviewer 1 Report

While this is a resubmission, short of some editing of the text, no real changes were made. The authors test several single markers, and state since they only found signicance with one, that would be the marker moving forward. However, their level of significance (0.043, 0.025) still does not meet multiple test correction (0.05/8 markers=0.007).

As another reviewer mentioned, single marker studies are not very informative and can often result in false positives. Coupled with the fact that this is a relatively small sample size in males only, this study doesn't seek to add much to an already exhaustive body of literature.

Author Response

ANSWER

Thank you for this comment. The text includes corrections suggested by the reviewers, for which we thank you very much - not only because the manuscript has become more clear and structured properly, but also because we were able to look at our results critically and discuss our group selection criteria, goals and conclusions.

We agree that Reviewer 2 passed critical comments to us, including those related to the selection of men only for the study group. Reviewer 2 is pleased with our explanations, which we are very happy and glad about.

And, as we have already said:

Our idea was to recruit males only, as they make up 80% of the patients in the centers and treat them as a homogeneous subgroup - not so much because of their genetic differences, but because of the type of addiction itself. This decision was made on the basis of our previous experience in recruiting a group of women and men. However, we have not left the women out - currently the recruitment process is underway (100 women addicted to alcohol and psychoactive substances). We hope to show analogous results concerning females soon. As for the fact that the users of substances were abstinent for at least 3 months and did not use  the active substance during the study, it was also intentional because the patients were staying in rehab centers with a total ban on consumption of the substance. For the duration of the study we chose a 3-month abstinence also because of the moment when psychological tests were being solved”.

The Reviewer writes:”……. that would be the marker moving forward. However, their level of significance (0.043, 0.025) still does not meet multiple test correction (0.05/8 markers=0.007)…..”  - it is with reference to Table 2. However, please note that we have made corrections to Table 4, where we looked for associations between addiction and mental disorders. This is where the Bonferroni correction was applied, and the genotyping frequency was only to serve as a guide for further study of these associations shown in Table 4. The Bonferroni multiple comparisons correction was applied (Table 3 and 4).

As for the comment: ”…. is a relatively small sample size in males only, this study doesn't seek to add much to an already exhaustive body of literature…..” the team is unanimous on the point that it is still a small group. However, 600 samples, studied and recruited carefully, contribute to the subject under study. This is a selected group and the study group was recruited in the addiction treatment centers and thoroughly examined in terms of clinical studies. It encouraged us to show the results of the study. But we keep on carrying out our studies. As already mentioned, the recruitment is continuous and a separate group will be made up of women (much limited in number though).

In the literature, despite many publications in regard to that research area, too little is known about the biology and genetics of addiction and existing associations. Our team still hopes to find more associations in addiction biology through research and compilation of different polymorphic variants of candidate genes and ultimately through the GWAS.

Reviewer 2 Report

>> In general, the authors have substantially improved the manuscript based on the reviewers’ initial reviews. Please find some further suggestions below the author’s responses to my initial review.

The genetic analyses the authors perform are slightly outdated. They investigate candidate genes for substance use; the field of genetics has now moved forward to the use of whole-genome data. Candidate gene studies have been largely unsuccessful due to our limited understanding of the genetic architecture of complex traits and they resulted in many false positive findings in the literature (Duncan et al., 2017).

Thank you for this comment. Bearing in mind the rich literature on the subject, we agree that our methods, especially those concerning molecular studies, are not the latest trend, but they are still used and are widely available. The choice of such methods was well thought over and, in a sense, made with a long-term view. As we have only about 600 samples, we decided to perform analysis as presented in this study, which shows associations in particularly selected and targeted genes. However, as the reviewer rightly notes, there are better methods. Our plan is to use them only for a short period of time. Recruitment and research on patients from addiction treatment centers is still ongoing. By planning one thousand samples, we will study the patients using GWAS. This is the right direction and we want to follow it. However, such analysis - which selects genes from the dopaminergic system seems valuable to us and showing associations - we hope that the results we present may already become a hint or the right track for other researchers.

>> The authors should discuss the weaknesses of the candidate gene design in their discussion. In this regards, I provided them with an important study (Duncan et al., 2017) that shows how candidate gene studies have been largely unsuccessful. At the moment they only included this reference for a different reason (as a reference to this sentence: “To advance treatment of these complex conditions, more research is needed to reveal biological mechanisms of mental disorders and polysubstance addiction vulnerability”).

Regarding the writing of the paper: the manuscript could use editing from a native speaker. Although the meaning of sentences is generally understandable, they are not all fluent and correct. Also, some parts are described in detail (such as the genotyping paragraph), whereas others need some more detail. For example the choice of 2 of the selected candidate genes has not been justified in the introduction and the paragraph about statistical analyses lacks details. Importantly, in the introduction the aim of the study can be described in a bit more detail.

Thank you for this comment. The manuscript has been, of course, re-edited by a native speaker. The comments regarding particular parts of the manuscript shown by the reviewer proved to be helpful to us and have been re-edited by us. The extended paragraph referring to statistical analysis has been supplemented with the information on discontinuation of the analysis of associations between the following gene polymorphisms: DRD2 rs1076560, DRD2 Tag1D rs1800498, DRD2Tag1B rs1079597, DRD2 Ex8 rs6276, DRD2 PROM. rs1799732, ANKK1 Tag1A rs1800497, DAT and mental disorders in people addicted to stimulants and the control group. The application of the Bonferroni multiple comparisons correction has also been described. Page 4, lines 169- 175.

>> The language of the manuscript has been improved substantially. The aim of the study has now been explicitly stated in the introduction and more details were provided about the statistical analyses.

Regarding the analyses: Did the authors correct for general covariates? Age, sex etc.? And when investigating polysubstance use including versus excluding stimulant use, did they check that the former group was addicted to more substances in general?

I think it would be good to provide a bit more details about the different study groups. I would also like a bit more information about the controls, are these healthy controls? How were they recruited? Are they matched regarding age and sex etc.?

Thank you for your comment. The controls were recruited as volunteers - the same tests as in the study group were performed on them and they were examined by a specialist psychiatrist. The control group, like the study group, was made up of men only, matched regarding age. In Table 1 the characteristics of multiple substance dependence, both in the group of addicts to stimulants and in other addicts, has been complemented. The justification in the abstract text has been complemented - paragraphs 19-22, 84-90, 324-329. The descriptions of the group have also been complemented - lines 176 through182 and 188 through 190.

>> The authors have satisfactorily addressed these points.

Minor comments:

The conclusions do not follow from the findings (neither in the abstract nor in the general text)

The conclusions have been complemented both in the summary (lines 36-50) and in the main body of the text 432-443.

In the abstract I miss the aim of the study

Thank you for your comment. Of course, the aim of the study was described in the summary (paragraphs 19-27).

Page 2, line 67: “… and environmental factors including personality features, depression and emotional distress [14,15,16,17].” The authors should realize that depression, personality traits and emotional distress are not solely environmental factors, but are also influenced by genetic factors.

Line 96: personality features, depression and emotional distress have been removed as environmental factors.

Table 1: “Median” should be N

The table has been corrected.

The authors have not corrected for multiple testing, maybe they should justify that in their manuscript.

The Bonferroni multiple comparisons correction - Table 3 and 4 - has been applied.

These amendments - lines 388- 402 are referred to in the text of the Discussion.

>> All minor comments have been satisfactorily addresses by the authors.

Author Response

ANSWER

Thank you for reviewing and analyzing our corrections and for the substantive comments that helped us improve the manuscript.

Reviewer 3 Report

To Authors:

The authors aimed to analyze polymorphisms of the genes (DRD4 Ex3, DRD2 (rs1076560, rs1800498, rs1079597, rs6276, rs1799732), ANKK1 Tag1A rs1800497,  DAT) in the group of patients diagnosed with polysubstance use disorder, including addiction to stimulants, and co-occurrence of specific mental disorders in the group of patients diagnosed with polysubstance use disorder, including addiction to stimulants, compared to the group of patients diagnosed with polysubstance use disorder, resulting in the outcomes that psychotic disorders were significantly more common in the examined group of males with polysubstance addiction including addiction to stimulants compared to the group of males with polysubstance addiction without addiction to stimulants, and different statistical significances were found in the frequency of the DRD4 Ex3 gene polymorphism: s / s is more common in the study group. The manuscript is largely well written and informative overall. However, there seem to be several major and minor concerns in this manuscript. The paper will be improved when the authors revise them according to the following comments:

[Major point]

All manuscript:

English sentences should be corrected and refined more throughout the manuscript.

[Minor points]

All manuscript:

Format and erroneous letters should be properly corrected and refined more throughout the manuscript.

All manuscript:

The authors should not use abbreviated words without explanation the first time they appear (e.g., ”Ex3” and “PROM”).

Author Response

ANSWER

Thank you for your reviews and comments. The whole text will be reviewed by a native speaker. The abbreviations for Exon 3 (Ex3) and the gene promoter (PROM) were explained when used for the first time in the text. Additionally, the DAT1 acronym was uniformly used throughout the manuscript.

Round 2

Reviewer 3 Report

To Authors:

The authors revised the manuscript according to my previous comments. The manuscript is largely well written and informative overall, so there seem to be no concerns in this manuscript. The paper may be accepted in the present form.

This manuscript is a resubmission of an earlier submission. The following is a list of the peer review reports and author responses from that submission.

Round 1

Reviewer 1 Report

Review. DRD4, DRD2, DAT1 and ANKK1 genes polymorphisms in patients with dual diagnosis of polysubstance addictions

In this manuscript the authors describe a study in which they compared the prevalence of certain polymorphisms in candidate genes between people addicted to multiple substances including stimulants, versus people addicted to multiple other psychoactive substances. They also investigate the presence of psychiatric disorders in people with polysubstance dependence versus a control group. They found that certain DRD4 polymorphisms are more common in the polysubstance group that uses stimulants versus the control group. Furthermore, they found that psychotic disorders occurred more frequently in the people addicted to stimulants.

The genetic analyses the authors perform are slightly outdated. They investigate candidate genes for substance use; the field of genetics has now moved forward to the use of whole-genome data. Candidate gene studies have been largely unsuccessful due to our limited understanding of the genetic architecture of complex traits and they resulted in many false positive findings in the literature (Duncan et al., 2017).

Regarding the writing of the paper: the manuscript could use editing from a native speaker. Although the meaning of sentences is generally understandable, they are not all fluent and correct. Also, some parts are described in detail (such as the genotyping paragraph), whereas others need some more detail. For example the choice of 2 of the selected candidate genes has not been justified in the introduction and the paragraph about statistical analyses lacks details. Importantly, in the introduction the aim of the study can be described in a bit more detail.

Regarding the analyses: Did the authors correct for general covariates? Age, sex etc? And when investigating polysubstance use including versus excluding stimulant use, did they check that the former group was addicted to more substances in general? I think it would be good to provide a bit more details about the different study groups. I would also like a bit more information about the controls, are these healthy controls? How were they recruited? Are they matched regarding age and sex etc?

Minor comments:

The conclusions do not follow from the findings (neither in the abstract nor in the general text)

In the abstract I miss the aim of the study

Page 2, line 67: “… and environmental factors including personality features, depression and emotional distress [14,15,16,17].” The authors should realise that depression, personality traits and emotional distress are not solely environmental factors, but are also influenced by genetic factors.

Tabel 1: “Median” should be N

The authors have not corrected for multiple testing, maybe they should justify that in their manuscript.

References

Duncan, L. E., Ostacher, M. & Ballon, J. How genome-wide association studies (GWAS) made traditional candidate gene studies obsolete. Neuropsychopharmacology : official publication of the American College of Neuropsychopharmacology 44, 1518-1523 (2019).

Reviewer 2 Report

Major comments:

This study was conducted using male patients only. There is no explanation for this choice (were females not available? were females analyzed but they were not significant? etc). None of the examined genes or disorders are sex-linked to warrant such a choice. Also, substance users had completed at least 3 months of abstinence and were not active substance users at the time of study. Results may be addressing such factors rather than chronic, toxic use of substance.

In addition, introduction states that DRD4S is 6 or fewer repeats, while DRD4L is 7 or more (lines 79-84). However, data was analyzed differently and DRD4S was considered 5 or fewer repeats, whereas DRD4L was 6 or greater (lines 154-155). This is a grave oversight and potentially indicates flawed study design/analysis. The main conclusion of the manuscript is that differences in short and long are significantly associated with study variables/outcomes.

None of the tests were corrected for multiple comparisons (such as a Bonferroni or FDR test), nor would any hold up to such tests. The authors fail to mention why they didn't do such corrections.

Minor comments: introduction and discussion sections could be improved with the addition of multiple paragraphs. Long blocks of text are hard to follow.